# The Cult of Our Lady of Fátima, Portuguese Colonialism, and Migration, c. 1930–c. 1980

Arpad von Klimo 

Department of History, The Catholic University of America, Washington, DC 20064, USA; klimo@cua.edu

**Abstract:** Emerging in 1917 amidst the anxieties of World War I, the Cult of Our Lady of Fátima began with a vision witnessed by three shepherd children in rural Portugal. News of the apparition quickly spread, drawing hundreds, then thousands of pilgrims to the Cova da Iria. This potent symbol of faith soon transcended its origin, migrating in the form of venerated statues and dedicated shrines that sprouted across the globe. Particularly intertwined with Portuguese emigration, the cult's reach extended to former colonies in Africa and Asia ("Ultramar") and distant communities like Brazil. Statues of Our Lady became beacons of familiarity and solace, offering "homes away from home" for displaced populations. This essay focuses on the discourses surrounding the cult between the early 1930s and 1950s, exploring how Fátima served as a focal point for navigating the social, political, and cultural conflicts inherent in the emigration experience.

**Keywords:** Marian cults; Catholicism; Portugal; migration; colonialism; Vatican

## 1. Introduction

"On 27 May [1955], four thousand Portuguese fishermen, clad in their colorful checkered shirts, walked fifteen to twenty abreast up the hilly streets of St John's to the Basilica of St John the Baptist, bearing the three-and-one-half-foot-high statue of *Our Lady of Fatima*, their gift of gratitude and friendship. Our Lady was to be a holy link between the two peoples, and, even today, worshippers pray before her in the specially prepared alcove to the left of the main altar. The six thousand men and women who filled the cathedral, along with those who lined the streets, were part of 'one of the most colorful, inspiring, and solemn events ever to take place in St. John's."[1] (Doel 2000, p. 43).

This impressive spectacle in the capital of Newfoundland is only one of countless examples of the ways a modern Catholic Marian cult, which had its origin in Portugal in 1917, became a specific religious practice that created a symbolic space in which emigrants and local communities could communicate a variety of meanings. The cult included ideas of belonging to a national community and its culture as well as its colonial or "civilizational" mission, but also attachment to a transnational religion. For most individuals and families, however, it was rather concerned about personal hopes and wishes, especially for emigrants and their specific situations.

Apart from these general observations, there is the long history of St. John, which could put our assumptions of "migration" into question: Although it was, in 1955, a place that was dominated by an English-speaking population of mostly Scottish decent, the Portuguese here did not fit into the usual narrative of the poor, Southern European immigrants who escaped their economically backward country to succeed in North America. In reality, St. John was probably one of the first Spanish or Portuguese settlements where fishermen caught cod. As such, we could ask ourselves whether the Portuguese who came later can be still described as "migrants." The story becomes even more complex when we take into account that the Portuguese fishermen were not immigrants to Canada but rather seasonal workers who were living in Portugal. A new approach, however, questions whether a strong distinction between the northern (British) and southern (Spanish/Portuguese) narratives

of European colonial expansion makes much sense for the 16th century, when various European activities were coordinated (Bouchard 2023). But, in any case, the procession and the statue of Our Lady of Fátima was understood as a symbol that had meaning for all the different communities in the town.

What was the meaning of the procession on 27 May 1955? The statue of Our Lady of Fátima was a gift Portuguese fishermen brought to the cathedral of St. John the Baptist, the Catholic mother church of the diocese of Newfoundland founded in 1855, which represented one of the largest Irish Catholic communities in North America. With the gift, the Portuguese fishermen wanted to express their gratitude for the hospitality shown by the population of St. John's. By leaving the statue inside the church, however, they also created a space in Newfoundland that held a strong meaning for Portuguese immigrants and that connected them to the Irish-Catholic community there. Finally, the statue of Our Lady of Fátima in the year 1955 could also be understood as a spiritual protection from communism in the context of the Cold War, six years after both Canada and Portugal were among the founding members of NATO. (Zimdars-Swarts and Morgan in: Margry 2021).

In my previous article in *Religions*, I focused on the political history of the Cult of Our Lady of Fátima. In this study, I describe the spread of the cult and its functions for the followers in the context of colonialism, Papal geopolitics, and most of all, in various forms of migration.

## 2. The Beginnings and the Establishment of the Cult in the 1920s and 1930s

Today, Fátima stands as one of the most popular destinations for Catholic pilgrims and tourists worldwide, drawing millions of visitors annually. What was once a field for grazing sheep, located approximately one hour north of Lisbon, has transformed into a significant pilgrimage and tourism site, including expansive churches, chapels, hospitals, hotels, restaurants, and shopping malls (Samiça 2018). Simultaneously, just a few decades after the apparitions, the cult of this Madonna—primarily represented through statues—had proliferated globally. How did this phenomenon unfold?

The success of this specific Marian cult, unlike many other apparitions that failed to gain church approval or develop into worldwide cults, can be attributed to at least five major developments, facilitated by various factors:

(1)  The unique circumstances faced by the predominantly rural population around Fátima and other parts of Portugal in 1917 played a pivotal role. During that period, they grappled with concerns about the preservation of their religious practices, threatened by the new republican government and its secularizing and anti-clerical political agenda (von Klimo 2022). Economic hardships among peasants, the anxieties stemming from Portugal's entry into World War I in 1916, existing Marian devotion, and awareness of the cult of Our Lady of Lourdes collectively fueled strong expectations and preparedness among thousands of believers to embrace the narratives of the three seer children and flock to the site immediately after news of the apparitions spread. Attempts to suppress the cult through violence, such as the arrest of the three children in August 1917 and the destruction of the chapel in 1919 by anti-clerical activists, only intensified devotion by adding miraculous occurrences, strengthening the resistance among the Catholic population against the new secularist regime.

(2)  From the early 1920s onward, Jose Alves Correia da Silva, the new bishop of Leiria (now Leiria-Fátima), ensured church control over the site by purchasing the land in 1920 and overseeing the construction of a basilica and a hospital in subsequent years. The bishop also initiated an investigation, concluding in 1930 with an announcement that the children's stories were "worthy of belief" (Madigan 2001, p. 58). After establishing close contact with the surviving visionary, Lucia dos Santos, the bishop encouraged her to document her memories from 1939 onward. These writings, including the "secrets" of Fátima, were translated into numerous languages, contributing to the cult's significance and popularity, especially in the context of Catholic anti-communism during the Cold War.

(3)    Another innovation by Bishop Da Silva was important for the popularization of the cult: In 1922, he inaugurated the newspaper *Voz da Fátima,* ("voice of Fátima"). (https://www.fatima.pt/pt/pages/voz-da-fatima, accessed on 2 January 2024). The paper has been published every month on the 13th, highlighting the days of the apparitions and providing an official, church-sanctioned narrative of the apparitions and everything related to them. It reported on events at the site or about the three children but also included articles about the history of the apparitions at Lourdes, as well as articles about Marian liturgy and similar topics: for example, about Padre Pio, an Italian monk famous for his hand stigmata. With the second issue (November 1922), *Voz da Fátima* started a series under the headline "Las curas de Fátima", with monthly articles that documented medical cases, photos of the patients, doctor's certificates, and other information on people who claimed they were cured with the help of Our Lady of Fátima. (*Voz de Fátima*, 13 January 1923). These healing stories strongly emphasized a practical aspect of visiting the shrine and of the cult in general: Our Lady of Fátima was there to help everyone who called her in their everyday struggles.

(4)    The shift to conservative governments in the 1920s and the establishment of Antonio Salazar's authoritarian regime (Estado Novo) in 1933 led to increased public support for the Catholic Church in Portugal and its cult. The "institutional regulation" of Fátima by the Church hierarchy integrated traditional, local practices of popular religiosity into a newly sanctioned domain of modern, individualizing religiosity. As Alfredo Teixeira writes, "the phenomenon of Fatima affirms itself in the field of detraditionalization of the religiousness of the Portuguese. This happens in two ways, which paradoxically call to a religious modernity: on one hand, a concentration on the narrative of the apparitions in privileging the doctrinal and ideological dimensions, in detriment to the miraculous plot; on the other hand, centralizing that message in a call to individual conversion, which accompanies the itineraries of individualization and subjectivity characteristic of that religious modernity" (Teixeira 2015, pp. 62–63). Fátima thus became a symbol of national relevance in this context.

(5)    Finally, since the early 1940s, the regime of Salazar and the cult received substantial backing from Pope Pius XII. The systematization and infrastructural buildup of the pilgrimage site by the local church, the political support from the Portuguese state, and the official endorsement from Rome all contributed to the establishment of the cult as the most important Catholic cult of Portugal with a strong aura that went far beyond the country.

### 3. Our Lady of Fátima in the Colonies: On a Divine Mission?

One significant aspect contributing to the expansion of the cult of Our Lady of Fátima beyond Portugal's borders since the mid-1920s was its association with Portuguese colonialism, backed by the support of missionaries and political regimes. Initially, the military dictatorship in 1926, followed by Salazar's Estado Novo from 1933 onwards, played crucial roles. Additionally, Pope Pius XII endorsed the cult and the "Ultramar" ideology.

In 1927, one of the earliest "Missions of Our Lady of Fátima" was established in Ganda, Angola, by Spiritan Fathers (Mission 2023). The Congregation of the Holy Spirit (Spiritans), founded in France in 1703, had been active in Angola since 1866, primarily in the field of education. Despite most of their missions being located in French colonies in Africa, this endeavor in Angola marked a notable collaboration with Portuguese activities. Although Portugal had established direct control over Angola only in the early 20th century, maintaining small coastal colonies since 1575, the Catholic missions played a vital role in the Portuguese colonial system. They provided education and hospitals, which the state could not have financed alone, aligning with the Catholic ideology of Portuguese colonialism (Cairo 2021).

The establishment of such missions should be viewed in the context of a significant political shift in Portugal. The brief military dictatorship in 1926 preceded the formation of Salazar's Estado Novo (1933–1975). Both regimes heavily relied on support from the

Catholic Church to implement their ambitious conservative programs for national renewal, particularly in culture and education. The objective was to reconstruct Portuguese society after the tumultuous period of the liberal republic (1910–1926). Within their policies, a focal point was the expansion and strengthening of control over the colonial empire. In the realm of colonial policy, Catholic missions were assigned a central role, particularly in education, seen as the primary foundation for bringing "civilization" and integrating all territories into the "Ultramar," the Portuguese empire (Madeira 2005).

The founding of the mission in Angola in 1926 was supposed to defend Portuguese interests against activities of "de-nationalization" in the form of two Protestant missions (from Switzerland and the USA) in the area. ("Missão de N. Senhora de Fátima em Africa", *Voz da Fátima*, 13 October 1930). In other words, spreading the cult of Our Lady of Fátima in the colonies was regarded as a powerful tool to strengthen their national Portuguese character and place within the colonial empire. This intention was also expressed in the *Estatuto Orgânico das Missões Católicas Portuguesas de África e Timor* (Organic Statute of the Portuguese Catholic Missions in Africa and Timor), published in the same year (Sánchez-Gómez 2009, p. 672).

In the following years, further laws and reforms were introduced, culminating in the integration of two major codes on the colonies in the constitution of the *Estado Novo* in 1933. The new legislation on colonies saw the Catholic missions as important "instruments of civilization and national influence" (Sánchez-Gómez 2009, p. 677). For many Catholic believers and the clergy, this even went so far as to understand the apparition of Fátima in 1917 as a sign of the divine mission of the Portuguese nation and its leader Antonio Salazar. More than 40 years later, in 1965, the Jesuit missionary Joaquim Guerra (1908–1993), who had spent many years in China and had founded an institute of higher education in Macao, where the cult of Our Lady of Fátima was installed in 1929, expressed this idea (see Bruxo 2004). When he was asked, during a university course, why the church supported Portuguese colonialism, Guerra referred to the privileged position of the church within the Estado Novo, saying that "the miracle was worked by our Lady of Fátima, to whom our country belongs. So, I said one day, when Sister Lucia [the surviving visionary of 1917, AvK], who was present, observed: 'Yes, but through Prof. Oliveira Salazar'. [...] *The Estado Novo is an awakening of the national conscience to the mission which God assigned to us, is Providence giving us for this purpose capacities and resources which we must exploit*" (Guerra n.d., p. 51. English quotation in: Ferreira 1974, pp. 68–69, my emphasis).

According to this idea, colonialism, and the Catholic missions in the colonies, was a major task the Mother of God had bestowed up Portugal when she had appeared in 1917! This was probably a popular idea, which explains why more and more colonial missions were dedicated to Our Lady of Fátima. Some examples from 1936 include the Benedictine Mission of Muchopes in the town of Mazucane in Mozambique, and another one in Moxico, Angola. ("Em Moçambique–Inauguração da nova escola de Nossa Senhora de Fátima, de Mazucane, sucursal da Missão de S. Benedito dos Muchopes", Voz da Fátima 1936, p. 3).

Finally, Pope Pius XII, who supported the regime of Salazar and hoped to counterbalance the influence of the Axis powers (Fascist Italy and Nazi Germany) on the country, expressed his full endorsement of Portuguese colonialism. Pope Pacelli also felt a close personal relationship to Our Lady of Fátima because he was consecrated archbishop at the Vatican on the 13th of May, 1917, the day of the first apparition of the Holy Virgin to the three shepherds.

The three most important documents in this context were the Concordat with Portugal, the Papal Encyclical *Saeculo Exeunte*, and the *Accordo Missionario,* all published in 1940. (The English translation of the concordat is published here: https://www.concordatwatch.eu/topic-38751.834, accessed on 2 January 2024) On the one hand, the concordat demonstrated that Salazar was not willing to give the church more influence on areas over which the Portuguese state had gained control since the 19[th] century—civil marriage, civil divorce, the appointment of bishops (nominated by the Vatican but with final approval from the government), as well as religious education in schools (remained voluntary). On the other

hand, the concordat strengthened the role of the church in general and symbolically (via its presence during state ceremonies, for example), but mostly in the colonies (Brandão 2004, p. 57).

In his Encyclical *Saeculo Exeunte Octavo*, Pope Pius XII praised the Portuguese colonial empire and its achievements and referred, as the first pope, to the Madonna of Fátima: "And indeed, the Catholic faith, which nourished the nation of Portugal from its very origin, was the principal force which raised your fatherland to the peak of its glory, extending the boundaries of both religion and empire. The Church adorned Portugal with all the embellishments of culture and rendered it worthy of its sacred endeavors in missions. [. . .] And now, when more than a few European nations have been lost to the Church because of the changes in these calamitous times, we see your people and their Spanish brothers opening paths and laboring for the Church in the spacious lands of Africa, Asia, and America. There they recruit numerous adherents of the Church to replace those who have miserably left her embrace. Then dioceses, parishes, sacred seminaries, monasteries, hospitals, and public orphanages arise almost everywhere in these places to prove the vital force and perennial virtue of the Catholic Church. [. . .] Those who have been called to the sacred orders of the contemplative life are to pray for this special intention, and the faithful, when *reciting the rosary so highly commended by the Blessed Virgin at Fátima, should entreat this same Virgin to intercede in favor of this divine vocation in order that the missions will flourish.*" (https://www.vatican.va/content/pius-xii/en/encyclicals/documents/hf_p-xii_enc_13061940_saeculo-exeunte-octavo.html, accessed on 19 July 2023, my emphasis).

With the backing of the papacy, the cult experienced further dissemination in the Portuguese colonial empire during the 1940s and the 1950s. The intention was to underscore the Portuguese national character of the "Ultramar," the envisioned community encompassing all areas under the country's control in Europe, Africa, and Asia. An illustration of this trend is evident in structures like the sanctuary of Nossa Senhora de Fátima in Namaacha, Province of Maputo, Mozambique. Constructed in commemoration of the 25th anniversary of the apparitions in 1942 (Neves 2016), the church was inaugurated two years later under the guidance of Cardinal Cerejeira, the Patriarch of Lisbon and a longtime associate of Salazar since their days at the University of Coimbra. The architectural style of the church, known as "estilo manuelino", a distinctly Portuguese Neo-Gothic/Renaissance style prevalent since the 16th century, was designed to accentuate its Portuguese national character. The building's style aimed to symbolize the continuity of Colonial Portugal from the outset of overseas expansion in the 15th century (with the first Portuguese outposts at the coast of Angola) through the 20th century, highlighting the identity between the "motherland" and the colony. This architectural style also conveyed the "ultramarine" ideology, asserting that Portugal and all its overseas colonies in Africa or Asia constituted integral parts of the "one and indivisible Portugal" (Brandão 2019, p. 222).

## 4. The "Migrating Statues" of Our Lady of Fátima after World War II and the Contradictory Papal Support of Portuguese Colonialism

Following the conclusion of World War II, numerous adherents of the cult held the belief that the Mother of God as Our Lady of Fátima played a major role in ending the war. In celebration of this belief and with the aim of expanding the cult to the Portuguese colonies and beyond, new statues of the Madonna were crafted and subsequently dispatched on "pilgrimages. The first such replica statue received special support from Pope Pius XII, who crowned her and bestowed the title "International Pilgrim Virgin Statue of Our Lady of Fátima" (Heitor 2019, examines how these statues were later utilized as a tool to promote tourism). In the summer of 1948, one of these statues departed from the harbor of Lisbon on a ship, embarking on a journey to Africa, to predominantly, but not exclusively, Portuguese colonies. The pilgrimage covered a route from the islands of São Tomé e Príncipe to Angola, then to Mozambique, visiting various South and East African colonies before ultimately reaching Cairo. In numerous locations along this route, the statue's visits served as inspiration for the construction of churches and sanctuaries dedicated to the cult.

As an illustration, when the statue reached Benguela (Angola) in July 1948, Governor Mario da Costa Ribeiro Zanatti (1898–1970) and priest Dom Manuel Junqueira committed to constructing a church for the Holy Virgin. However, it took an additional 20 years until the cathedral was inaugurated (https://www.verangola.net/va/pt/012020/sugestoes/17 799/Catedral-de-Benguela-O-que-fazer.htm, accessed on 19 July 2023).

This example highlights the active involvement of the colonial elite in propagating the cult, without dismissing its potential popularity among the indigenous population over time. In the Mozambican town of Namaacha, the annual procession of Our Lady of Fátima continues to attract thousands of believers today, as evidenced by a photo from the 2019 procession (https://clubofmozambique.com/news/this-saturday-more-than-15000-catholics-to-make-the-pilgrimage-to-namaacha/, accessed on 19 July 2023). A brief video capturing the 2017 procession in Namaacha during the 100th anniversary of the apparitions is available here: (https://www.youtube.com/watch?v=aqHA9rc6CYA, accessed on 19 July 2023).

Ultimately, the Portuguese influence evident in the church's construction could not prevent the cult from taking root in other, non-Portuguese cultures. The procession serves as an example of the Africanization of Mozambican Catholicism (Morier-Genoud 2019). Didier Péclard argued that the institutions of the Catholic Church in Angola both fostered and hindered the growth of Angolan anti-colonial nationalism (Péclard 1998, p. 160). The missionary work, although ambivalent, eventually led to protests against the colonial regime, notably by the Spiritan fathers, who publicly opposed the suppressive policies of the Portuguese regime in Angola in a 1970 open letter (Péclard 1998, p. 169). Additionally, the Africanization of the clergy, initiated by the Vatican since at least 1945, was articulated in Saeculo Exeunte, where Pope Pius XII prayed "that God may inspire both the people of Portugal and those of the nations subject to your rule to become priests or coadjutor brothers or nuns or catechists devoted to missionary work" (https://www.vatican.va/content/pius-xii/en/encyclicals/documents/hf_p-xii_enc_13061940_saeculo-exeunte-octavo.html, accessed on 15 September 2022, my own emphasis). While Pope Pius XII had staunchly supported Portuguese colonialism, the Vatican's practice to promote indigenous priests undermined the Portuguese government's attempts at a "nationalization of missionary activity" in the colonies in the long run, evolving into, according to Pedro Ramos Brandão, "a constant source of friction with the Catholic Church" (Brandão 2019, p. 223).

Lopes and Santos, however, argue that the "euro-centrism" of the Portuguese Catholic Church continued after the breakdown of the colonial empire in 1975 in the form of institutions and agents related to the center of the church in Rome and because of the missionary activities outside of Europe were rooted in the history of European expansion and colonialism (Lopes and Santos 2000). The authors include in this the colonial/post-colonial situation of about 840 Portuguese missionaries, of which 54% were female (data from 2000), active in 390 missions, of which 67% were located in Africa (mostly former Portuguese colonies like Mozambique and Angola) and 22.7% in Latin America (16% in Brazil, and a further 6.7% in other countries).

## 5. Portuguese Migration Patterns to and beyond the Colonies, 1920s to 1970s

What effect did the Salazar regime's ideology of the "Ultramar", the "unity" between Portugal and its colonies (since 1951: "overseas provinces"), have on Portuguese migration? In the 19th century and the first half of the 20th century, Portuguese emigration to the colonies was relatively weak, compared to the main destinations of Brazil or North America (Morier-Genoud 2012). The regime of Salazar did not encourage migration to the colonies, because it was afraid of a "Brazilian effect", meaning a Creole movement for autonomy or independence from the mother country. Only after World War II did the regime begin to cautiously support the move to the colonies in order to strengthen the European elements there. Migration to the "overseas provinces" in Africa would peak around 1965. For many emigrants, it was a way to leave the poor, rural areas; they were hoping to settle in urban

environments in the colonies since that was sometimes easier than moving to a larger city in Portugal itself.

Since the 1970s, the migration patterns changed and more people of European, African, and Asian (mostly Indian) background moved from the colonies to Portugal instead the other way round. Many of these various diasporas had helped to maintain the Portuguese Empire. Portugal itself had become a country of mass emigration to Western Europe, particularly to France, Belgium, West Germany, Switzerland, and Luxembourg.

For the Portuguese diaspora in France, the cult had been central as a practice that helped the migrants to bridge the differences between the two countries and their cultures (Volovitch-Tavares 2009). Their number rose from around 10,000 in 1921 to 54,000 in 1951 and exploded to reach 750,000 in 1975, becoming the largest group of immigrants in France (numbers: Volovitch-Tavares 2009, p. 14).

In the rural areas where they came from, the rhythms of life were organized around religious practices, although not around theological doctrines, while in France, they found themselves in urban, industrial, secularized areas. Even the Catholicism that was prevalent in France after the Second Vatican Council was very different from the more conservative, pre-conciliar variant that was prevalent in the rural areas of Portugal. Therefore, the cult of the Holy Virgin with its rather magical practices, like the miraculous healing of diseases or the solving of financial troubles, became the central place in which these differences could be temporarily overlooked. This explains why the statue of Our Lady of Fátima has populated many houses and churches in the emigration countries.

The sociologist, anthropologist, and former Portuguese missionary Policarpo Lopes provided a profound study of the practices of Portuguese immigrants in Belgium during the 1980s, based on a large number of interviews and participant observations (Lopes 1992). He also interpreted Fátima as a popular religious symbol that helped to master the situation created by migration. According to Lopes, the cult fulfilled various functions for the immigrants and their families. Most of all, it eased the tensions created by their specific transitory or liminal situation. The liminal place of the immigrant locates her between two countries, two cultures, and two languages. In the case of most Portuguese immigrants in Belgium, they were caught between a rural, traditional, conservative community of origin and an urban, industrial, dynamic, modern, individualistic, consumerist society of migration. They were also caught between an older generation that was shaped by the traditional background and the younger generation, which was more integrated into the new, modern society and its culture. Other scholars have emphasized the significance of the cult, and the pilgrimage especially, as a tool of empowerment specifically for women in difficult situations (Hermkens et al. 2009, pp. 8–15).

The Holy Virgin is a motherly figure that is related to the old country: the communal everyday practices of popular religiosity with its more pragmatic, symbolic, magic features in contrast to the reality in the new, modern, foreign society. Praying with her gives comfort, hope, and smooths over contradictions and conflicts, because she can easily be integrated into the new environment, as a statue in the home and in the chapel or church where she waits for the believers. And Our Lady of Fátima can even be much more, at least for certain believers, as this statement demonstrates:

"It is Her whom I owe my life. When I had the Asian Influenza in Portugal, they gave me a medicine against which I was allergic which I didn't know at the time. I spent three months in bed and had not the strength to get up. I felt very bad and I couldn't speak or see. It was the 12th of May. . . they (my family) went to find a place at my uncle where they could follow the ceremonies at Fátima. After the second decade [of the Rosary], I got up and continue to pray with them. Three doctors came and we told them what happened: I and the whole world believes that this is a miracle. The next day I could march as I never could before. Therefore, I am very devoted to Our Lady of Fátima." (cit. in Lopes 1992, p. 241, my own translation).

The Holy Virgin of Fátima, in this story, overcomes the forces of death (see also Gemzöe 2009) Such miracles, which are also often fulfilments of the wishes that believers have made

during prayer from the holy Mother of God, are "paid back" by visiting the shrine in Fátima. Such visits to "give back" or to thank for a favor granted by Our Lady have been a very common practice for many Portuguese immigrants. At the same time, such visits, mostly during the summer holidays, also function as bridges between the modern, urban, individualistic society of migration and the old, rural country of origin with its close-knit communities: Portuguese families would pray at the shrine in Fátima in the morning and go to a beach in the afternoon or have lavish dinners in the evening, thus combining the religious practices of the old community with the new ways of modern tourism. The growing touristic ensemble and services available around the shrine since the 1950s tell the story of this compromise between different places which bridge the deterritorialization of the migrants, at least for a moment, and eases the tensions between communal, traditional origins and new, more individualistic, modern consumer society.

## 6. Conclusions

In Portugal, the Cult of Our Lady of Fátima evolved during a period marked by political, economic, and cultural turmoil. It was established by a Catholic Church aligning with an authoritarian regime characterized by a conservative, nationalist, and colonialist agenda. However, this political dimension was superimposed onto a religious practice that had already garnered the devotion of hundreds of thousands of believers, serving as a source of private solace, relief, and comfort. Many adherents also held a belief in the cult's miraculous healing powers for both the body and the soul. The cult's multifunctionality, serving as both a national religious symbol of Portugal and a potent resource for personal religious devotion, elucidates why it persisted beyond the authoritarian regime. Instead, it underwent a transformation into a contemporary popular religious practice that seamlessly integrated with the dynamics of tourism and other facets of the consumer society prevalent in the late 20th and early 21st centuries. The act of visiting the shrine in Fátima and installing statues of Our Lady of Fátima in local churches and homes has offered (and continues to offer) a valuable means to bridge the deterritorialization and alienation experienced by immigrants. In essence, prayer and the presence of the statue create a temporal home away from home, establishing a connection for immigrants with their cultural roots.

**Funding:** This research received no external funding.

**Institutional Review Board Statement:** Not applicable.

**Informed Consent Statement:** Not applicable.

**Data Availability Statement:** Data are contained within the article.

**Conflicts of Interest:** The authors declare no conflict of interest.

## Note

1. A short video of the procession, broadcasted by Canadian state television can be watched here: https://www.cbc.ca/news/canada/newfoundland-labrador/connection-between-portuguese-fishermen-basilicas-fatima-statues-1.3612352 (accessed on 1 August 2023).

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
