# Peer review of "The Cult of Our Lady of Fátima, Portuguese Colonialism, and Migration, c. 1930–c. 1980"

_religions, doi:10.3390/rel15030255_

Round 1

Reviewer 1 Report

Comments and Suggestions for Authors

I enjoyed reading this article. Examining the evolution of the cult of Our Lady of Fatima, the author skillfully analyses a range of political, economical and cultural factors influencing it. The article also raises some intriguing questions about migration. It is very well researched and written with clarity and effectiveness.

I have just one request for clarification. The author mentions the "rather magic practices" of the cult of the Holy Virgin (line 312) and "magic features" (line 330). I suggest that the readers would benefit from a brief explanation and a couple of examples of these practices and features. 

Thank you! 

Author Response

Reviewer's question:

The author mentions the "rather magic practices" of the cult of the Holy Virgin (line 312) and "magic features" (line 330). I suggest that the readers would benefit from a brief explanation and a couple of examples of these practices and features. 

Author's response: (This is a very good point! I'm trying to describe the differences between rural Catholicism in Portugal and urban Catholicism in France by focusing on the cult of Our Lady of Fatima. What does "magic practice" mean in this context? 

I think a good  example are the healing "miracles" related to Our Lady of Fatima which would have been regarded with more skepticism in urban France compared to rural Portugal (not of course in Lourdes!). I will add a sentence about these miracles which can be interpreted as magical practices.

So, I changed the sentence to:

"Therefore, the cult of the Holy Virgin with its rather magical practices, like the healing of diseases or the solving of financial troubles, became the central place in which these differences could be, temporarily, overlooked." 

I hope this clarifies the point.

Reviewer 2 Report

Comments and Suggestions for Authors

Interesting article, focusing on Our Lady of Fatima, displacement and migration, especially in relation to Portugal. In addition to a few editorial issues, see below, the article would benefit from a more thorough engagement with literature on Marian devotion, and Fatima in particular, and the interplay between migration and Marian devotion. I miss references to various contributions in Margry's edited volume on Cold War Mary, and in Hermkens'et al Moved by Mary, which has two chapters on the International Pilgrim Virgin statue. Also Tweed's work on Diasporic religion (Our Lady of Exile) should be included to engage in more depth with the interplay between migration and Marian devotion. At the moment, the paper lacks analytical depth and engagement with relevant academic studies. While the information provided is interesting, the argument could/ should be improved by adding more academic engagement and thereby depth and interest.

I have highlighted some of the editorial issues in the text, see attachment.

Author Response

Reviewer 2 asked for more analytical depth and an engagement with the existing academic literature, particularly three books on Marian Devotion/pilgrimage and migration/exile. I have borrowed the three books from our library and will add some of the findings of these books, and add some more analytical insights. 

On line 96-111, I have included one new paragraph that emphasizes the role of healing stories which gives more depth to the story of the establishment of the cult. 

"Another innovation by Bishop Da Silva was important for the popularization of the cult: In 1922, he inaugurated the newspaper Voz da Fátima, (“voice of Fatima”). (https://www.fatima.pt/pt/pages/voz-da-fatima). The paper would appear every month, on the 13th, highlighting the days of the apparitions, providing an official, church-sanctioned narrative of the apparitions and everything related to them. It reported on events at the site or about the three children but included also articles about the history of the apparitions at Lourdes, as well as articles about Marian liturgy and similar topics, for example about Padre Pio, an Italian monk famous for his hand stigmata. With the second number (November 1922), Voz da Fátima started a series under the headline “Las curas de Fátima”, with monthly articles that documented medical cases, photos of the patients, doctor’s certificates, etc and other information on people who claimed they were cured with the help of Our Lady of Fatima. Voz de Fátima, 13 January 1923. VF0080_1923-01-13.pdf (fatima.pt). These healing stories strongly emphasized a practical aspect of visiting the shrine and of the cult in general: Our Lady of Fatima was there to help everyone who called her in their everyday struggles."

On lines 349-51, I have added one sentence with a reference on one of the three books the reviewer suggested:

"Other scholars have emphasized the significance of the cult and the pilgrimage especially as a tool of empowerment specifically for women in difficult situations. (, 8-15)" 

I also have added two important articles: 

  1. Gemzöe, Lena. “Caring for Others. Mary, Death, and the Feminization of Religion in Portugal”, In: Moved by Mary. The Power of Pilgrimage in the Modern World. Ed. by Hermkens, Anna-Karina; Jansen, Willy; Notermans, Catrien, Ashgate, 2009, 149-163.
  2. Hermkens, Anna-Karina; Jansen, Willy; Notermans, Catrien. Introduction. In: Moved by Mary. The Power of Pilgrimage in the Modern World. Ed. by idem, Ashgate, 2009, 1-15.

However, I didn't find the other book the reviewer recommended, Our Lady of the Exile (on a Marian cult among the Cuban diaspora in Miami) very helpful since I have focused on the Portuguese case here.